# Robust Kalman Filter Soil Moisture Inversion Model Using GPS SNR Data—A Dual-Band Data Fusion Approach

**Lili Jing [1], Lei Yang [2,3,*], Wentao Yang [4], Tianhe Xu [1], Fan Gao [1], Yilin Lu [5], Bo Sun [2], Dongkai Yang [3], Xuebao Hong [3], Nazi Wang [1], Hongliang Ruan [6] and José Darrozes [7]**

[1] Institute of Space Science, Shandong University, Weihai 264209, China; 202121037@mail.sdu.edu.cn (L.J.); thxu@sdu.edu.cn (T.X.); gaofan@sdu.edu.cn (F.G.); wnz@sdu.edu.cn (N.W.)

[2] College of Information Science and Engineering, Shandong Agricultural University, Tai'an 271018, China; sunb@sdau.edu.cn

[3] School of Electronic and Information Engineering, Beihang University, Beijing 100191, China; edkyang@buaa.edu.cn (D.Y.); joyce_hong2008@buaa.edu.cn (X.H.)

[4] School of Geological Engineering and Surveying and Mapping, Chang'an University, Xi'an 710054, China; 2019226007@chd.edu.cn

[5] China Association of Remote Sensing Application, Beijing 100094, China; luyilin@cresda.com

[6] Business School, Jinhua Polytechnic, Jinhua 321000, China; 20121011@jhc.edu.cn

[7] Laboratoire Géosciences Environnement Toulouse, Université Paul Sabatier, 31400 Toulouse, France; Jose.DARROZES@Get.omp.eu

\* Correspondence: yanglei@sdau.edu.cn

**Abstract:** This article aims to attempt to increase the number of satellites that can be used for monitoring soil moisture to obtain more precise results using GNSS-IR (Global Navigation Satellite System-Interferometric Reflectometry) technology to estimate soil moisture. We introduce a soil moisture inversion model by using GPS SNR (Signal-to-Noise Ratio) data and propose a novel Robust Kalman Filter soil moisture inversion model based on that. We validate our models on a data set collected at Lamasquère, France. This paper also compares the precision of the Robust Kalman Filter model with the conventional linear regression method and robust regression model in three different scenarios: (1) single-band univariate regression, by using only one observable feature such as frequency, amplitude, or phase; (2) dual-band data fusion univariate regression; and (3) dual-band data fusion multivariate regression. First, the proposed models achieve higher accuracy than the conventional method for single-band univariate regression, especially by using the phase as the input feature. Second, dual-band univariate data fusion achieves higher accuracy than single-band and the result of the Robust Kalman Filter model correlates better to the in situ measurement. Third, multivariate variable fusion improves the accuracy for both models, but the Robust Kalman Filter model achieves better improvement. Overall, the Robust Kalman Filter model shows better results in all the scenarios.

**Keywords:** GNSS; Signal-to-Noise Ratio; soil moisture; Robust Kalman Filter; data fusion

## 1. Introduction

Soil moisture is an essential component in the continental water–carbon cycle, and a key parameter for quantifying the energy and water exchange between land surface and atmosphere [1–4]. Precise soil moisture monitoring is an important foundation for achieving high yields of agriculture production. L-band microwaves have significant advantages in soil moisture remote sensing, for being little affected by clouds and atmosphere, and for its ability to penetrate vegetation. GNSS constellations can provide massive L-band signal sources for free [5–7]; a novel microwave remote sensing approach, GNSS-R (Global Navigation Satellite System-Reflectometry) technology was proposed by M. Martin-Neira in 1993 [8], which typically uses an up-looking antenna for receiving direct signals from GNSS satellites, and two down-looking antennas for receiving LHCP (left-hand circular

polarization) and RHCP (right-hand circular polarization) reflected signals from the Earth's surface. At first this method was mainly used for sea-states monitoring [9], and thereafter for land parameter observations such as soil moisture [10–13]. In 2008, K. Larson developed a new method called GNSS-IR [14], which uses a single RHCP antenna to receive both the direct and reflected signal simultaneously, with a Geodetic GNSS receiver to record the interference of the two signals in the SNR (Signal-to-Noise Ratio) file. Later, Zavorotny and Larson explained the GNSS-IR theoretically by developing a physical model of the interference mechanism together [15]. Compared to the conventional GNSS-R technology, GNSS-IR uses off-the-shelf commercial receivers and antennas, so that it has a great merit of low-cost, which makes it easier to build large scale in situ monitoring systems for soil moisture measurement.

Nievinski studied the forward model of GPS multipath signal for near-surface reflectometry thoroughly [16], and later built an open-source GPS multipath simulator in Matlab/Octave for demonstrating the impact of environmental observables such as soil moisture and snow density [17]. However, as the land surface is complex and varied, simulators may not fully simulate the real situation. Chew proposed a method for determining whether SNR data are significantly corrupted by vegetation and for correcting these effects [18]. Recently, leading GNSS-R groups are only just starting to grapple with the various complexities of land GNSS-R reflections and developing complementary models [19–21]. In addition, there are scholars that have combined remote sensing technology and GNSS-IR technology in estimating vegetation water content [22]. Han proposed a method to reconstruct direct and multipath signal from SNR data and then calculate the dielectric constant of soil [23]. Ting Yang retrieved the dielectric constant from BDS (BeiDou Navigation Satellite System) SNR data by using an analytical model and verified the applicability though experimentation [24].

R. E. Kalman introduced his famous discrete data filtering technique in 1960 [25]. The Kalman Filter algorithm has been extensively applied in GNSS [22], INS [26], data assimilation [27] and many other research fields for its ability to provide an efficient way of computing the least squares problem using a recursive method. As of recent years, the Kalman Filter has been applied in GNSS-R sea level monitoring [28] and sea wind retrieval [29–31], but little research has been conducted regarding soil moisture inversion.

The reflected signal of different frequency carriers has different information than the reflected surface, therefore the data fusion algorithm was introduced to GNSS-R applications. Nazi Wang proposed a Sea Level Estimation method base on GNSS dual-band carrier phase linear combinations and achieved altimetric accuracy <0.2 m. Wang et al. used a peak weighting method to fuse GPS L1 and L2 SNR for wind speed retrieval [32]. Most of the existing GNSS-IR soil moisture retrieval methods are focused on building an empirical model and only use a single-band of GNSS signal. However, a multi-band data fusion algorithm for GNSS-IR soil moisture measurement has rarely been studied directly. Therefore, in this study we have established a Robust Kalman Filter algorithm to retrieve soil moisture and to improve the robustness and accuracy of the retrieval.

This article aims to examine and evaluate the potential of data fusion algorithms for GNSS-R applications. First, an author-proposed robust regression model will be introduced, as well as an optimized novel Robust Kalman Filter model. For validation, a dataset collected at Lamasquère (South France) will be analyzed, and the third section presents the results obtained with both robust regression model and Robust Kalman Filter model. These model retrievals are compared to (1) a classical model obtain using an empirical regression model and to (2) measurements of in situ Theta Probe ML3 sensor. The last section concludes this study by highlighting pros and cons of the different methods.

## 2. Methods

GNSS reflectometry works like Bi-static Radar systems, which consider the GNSS satellites and receiver as the radar transmitter and receiver, respectively. The main idea of GNSS-IR technology is to use a single RHCP antenna to receive both the direct and

reflected signal simultaneously [11]. In the scenario of in situ observation with low antenna height (the reflected over-path must be less than L1 C/A code wavelength ~293 m), the difference of Doppler shift between direct and reflected signal can be ignored because our application scenario is ground based measurement. If it is being used with space-borne GNSS-R receivers, the Doppler shift cannot be ignored. At the same time, if the transmit path's difference is less than one chip, so that the two signals are coherent, then the two signals are interfering each other at the center of antenna. The scenario of interference generation is demonstrated in Figure 1.

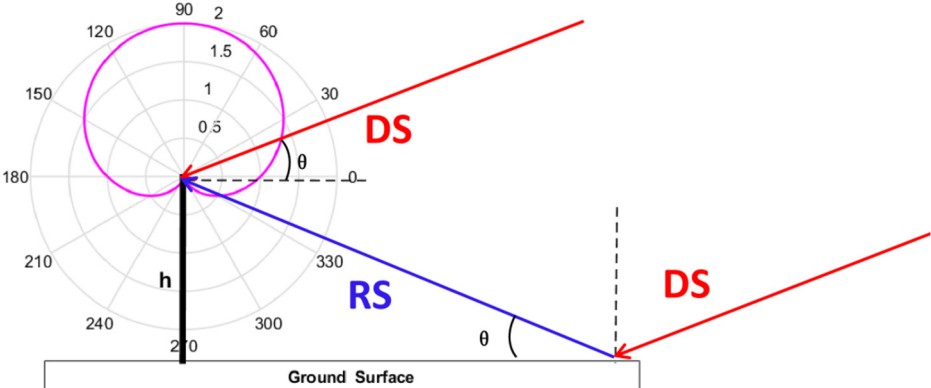

**Figure 1.** Scenario of Interference Generating of GNSS-IR, where $\theta$ is the elevation angle of the considered GNSS satellite, blue line represents the antenna gain pattern, and $h$ is the antenna effective height. DS is the direct signal and RS the reflected one.

The reflected signal contains more RHCP component in the low elevation angle scenario, so that the interference is more significant in that case. The interference phenomenon is recorded in the SNR data stored by the geodetic GNSS receiver. Following [16,17], SNR can be expressed as a combination of direct and reflected signal as shown in Equation (1):

$$SNR^2 = A_d^2(\theta) + A_r^2(\theta) + 2A_d(\theta)A_r(\theta)\cos(\varphi) \tag{1}$$

where $A_d$, $A_r$ is the amplitude of direct and reflected signal, respectively, $\varphi$ is the phase difference between reflected and direct signal, and $\theta$ is the elevation angle of GNSS satellite. When the elevation angle $\theta$ changes, the phase difference $\varphi$ also changes, by which the oscillation in the SNR amplitude is created.

GNSS-IR technology normally utilizes geodetic receivers and antennas. In that case, the antenna's gain pattern is designed to suppress the multipath signal coming from the bottom side for better positioning accuracy, i.e., the amplitude of SNR is mainly contributed by the direct signal (see antenna gain pattern, Figure 1).

Supposing $\delta$ is the difference between transmit paths of the reflected signal and direct signal, then $\varphi$ can be expressed as Equation (2):

$$\varphi = \frac{2\pi}{\lambda}\delta = \frac{4\pi h}{\lambda}\sin\theta \tag{2}$$

where $h$ is the effective height of the antenna, which refers to the distance from antenna phase center to the reflecting plane, and $\lambda$ is the wavelength of GNSS carrier wave.

Due to the microwave that penetrates the soil a few centimeters or decimeters (depending on the ground composition and the soil moisture) when reflected by soil surface, the reflecting plane is beneath the soil surface, so the effective antenna height varies in a range of few centimeters/decimeters with respect to the soil moisture, a key parameter in determining the penetration depth of the microwave. As the moisture usually does not change much in a few hours without precipitation, the effective antenna height can be treated as a constant to simplify the analysis.

The modified oscillation frequency can be derived by Equation (3):

$$f_x = \frac{d\varphi}{d(\sin\theta)} = \frac{4\pi h}{\lambda} \tag{3}$$

For soil moisture measurement the direct signal is not of interest and typically can be removed by a second-order polynomial fitting and then derive the reflected component only, which refers to the multipath component that can be expressed by Equation (4):

$$SNR_m(\theta,\varepsilon) = A_m \cos\left(\frac{4\pi h}{\lambda}\sin\theta + \phi(\theta,\varepsilon)\right) \tag{4}$$

where $\varepsilon$ is the dielectric constant of soil, $A_m$ and $\phi$ are the amplitude and phase of multipath oscillation, respectively, and $h$ is the effective antenna height. Typical data processing method is deriving the frequency $f_x$ and $h$ by using Lomb–Scargle Spectral Analysis (LSSA), and then deriving $A_m$ and $\phi$ by using least mean square estimation, which corresponds to a conventional empirical model used to retrieve soil moisture.

Existing studies show that normally $A_m$ and $\phi$ have higher correlation to soil moisture than $f_x$ and $h$ [13,14], so in this study we use $A_m$ and $\phi$ to build soil moisture inversion models. Here, we define $j$th day amplitude and phase observable on L$k$ band as $a_j^k$, $p_j^k$ ($k = (1, 2)$), and then the observables' time series vectors are $A^k$, $P^k$ which are constituted by daily observables $a_j^k$, $p_j^k$, respectively. In the same way the soil moisture time series vector can be defined as $Y \in R_+^N$ which is constituted by daily soil moisture $y_j$. In general, a GNSS-IR soil moisture retrieve model is a map function from $A^k$, $P^k$ to $Y$.

In this section, the conventional linear regression model and an author-proposed robust regression model will be introduced. Furthermore, a novel Kalman Filter model will be proposed for soil moisture inversion.

### 2.1. Conventional Linear Regression Soil Moisture Inversion Model

The general idea of conventional linear regression is to build a map from independent variable to dependent variable matrix, as shown as Equation (5):

$$Y = \beta X + v \tag{5}$$

where $\beta$ is different coefficients being calculated for each satellite at each frequency band and $v$ is the residual vector. $Y$ is the soil moisture time series for the whole measuring period. $X$ is observation or its combination.

In this study we use three different strategies to make the map for GNSS-IR linear regression.

In the first strategy, we used a single-band univariate linear regression. In this case, $X$ is the single vector $A^k$ or $P^k$. In the second strategy we used dual-bands data fusion univariate linear regression; in this case we can define two matrices $A = [A^1, A^2]$, $P = [P^1, P^2]$ and $X$ can be either $A$ or $P$. In the last strategy we used dual-bands data fusion multivariate linear regression. In this case we build a joint observable matrix $\begin{bmatrix} A & P \end{bmatrix}$.

In the classical process of linear regression, the least mean square method is generally used to determine $\beta$ and $v$, but the disturbance of outlier noise cannot be effectively eliminated. Therefore, a robust regression model and a Robust Kalman Filter model are proposed to solve this issue.

### 2.2. Robust Regression Soil Moisture Inversion Model

In thesis [33], the authors proposed a robust regression soil moisture inversion model (in Chinese). For convenience, we present now a brief description of this method in this study.

Massive studies and experimental campaigns show that the soil moisture has a near-linear relation to the observables of GNSS SNR multipath oscillation. Regression is a

statistical method to build the relationship between the independent variables and the dependent variables. Robust regression is a proper algorithm to suppress the effect of the environmental noise and the thermal noise induced by GNSS receiver and antenna.

The general form of regression model is shown as Equation (5) in Section 2.1. The residual vector $v$ of the equation includes the constant bias of the linear regression function and also random noise.

The concept of robust regression is based on M-estimate by utilizing the Iterative Reweighed Least Squares (IRLS) method for regression coefficient estimating [34].

A function $Q$ is defined as Equation (6):

$$Q = \sum_{j=1}^{N} g(v_j) \tag{6}$$

where $g(v_j)$ is the Huber robust error function defined as Equation (7):

$$g(v_j) = \begin{cases} \frac{1}{2}(v_j)^2 & if |v_j| \leq c \\ c|v_j| - \frac{1}{2}c^2 & if |v_j| > c \end{cases} \tag{7}$$

where $c$ is harmonic coefficient, which generally is an empirical value. According to article [35], when $c$ is set to 1.345 the regression can achieve 95% efficiency with high robustness, and this value is also the default value of Matlab function robustfit, so in this study we also set $c$ is to 1.345.

Then, the regression problem turns to an M optimization problem defined as Equation (8) for each frequency:

$$\begin{aligned} & \underset{H_j \in \Re^N}{\text{argmin}} Q, \\ & s.t. v = Y - \beta X \end{aligned} \tag{8}$$

when $Q$ is minimalized, $\forall \partial Q / \partial \beta = 0$, then Equation (9) is denoted:

$$\sum_{j=1}^{n} f(y_j - \sum x_j \beta_j) x_j = 0 \tag{9}$$

where $f(v_j) = dg(v_j)/dv_j$, then Equation (10) is denoted:

$$f(v_j) = \begin{cases} -c & if \ v_j \leq c \\ v & if \ v_j \leq c \\ c & if \ v_j \leq c \end{cases} \tag{10}$$

For achieving higher robustness, a scaled estimation value $s_j$ for each satellite is introduced to standardize the residuals. Following [34], we obtain Equation (11):

$$s_j = \frac{med|v_j - med(v_j)|}{0.6745} \tag{11}$$

where 0.6745 is median absolute deviation proposed by Hample [36] to guarantee the unbiased estimation under normal distribution, and $med(a)$ is the for denoting the median absolute deviation. Therefore, we can normalize the residuals as Equation (12):

$$\mu_j = v_j / s_j \tag{12}$$

The weight of the observation on $j$th day can be defined as Equation (13):

$$
\begin{aligned}
W_j &= \frac{f(\mu_j)}{\mu_j} \\
\boldsymbol{W} &= diag[W_j] \\
\boldsymbol{X}^T \boldsymbol{W}(\boldsymbol{Y} - \boldsymbol{\beta X}) &= 0
\end{aligned}
\tag{13}
$$

Then, we can obtain the iteration formulation of robust regression defined as Equation (14):

$$
\widetilde{\boldsymbol{\beta}} = \left( \boldsymbol{X}^T \boldsymbol{W X} \right)^{-1} \boldsymbol{X}^T \boldsymbol{W Y}
\tag{14}
$$

The main idea of the robust regression model is to assert different weights for different points depending on its residual—the smaller the residual of one point, the greater weight it will have. Then, the weights are optimized by iterating a weighted algorithm for 15 times and the first three observables are not used.

### 2.3. Robust Kalman Filter Soil Moisture Inversion Model

In our second model we developed a processing chain using a Kalman Filter. The idea of a Kalman Filter is to use recursion of input and output values to calculate and update by least mean square error estimation of the state. In this section we establish the state equation and observation equation. Then, we need to ameliorate the observation equation using the Huber-M estimation method. The standard Kalman Filter model assumes the true state at time $j$ is evolved from the states at time $j - 1$ [26], as shown in Equation (15). At time $j$, daily soil moisture $y_j$ of the true state $x_j(a_j, p_j)$ is made as Equation (16):

$$
x_j = \boldsymbol{F}_j x_{j-1} + \omega_j
\tag{15}
$$

$$
y_j = \boldsymbol{H}_j x_j + \varepsilon_j
\tag{16}
$$

where $\boldsymbol{F}_j$ is the state transition matrix which is applied to the previous state $x_{j-1}$; $\omega_j$ is the process noise with a value of zero which is assumed to be drawn from a zero-mean multivariate Gaussian distribution $\chi$, with covariance $\boldsymbol{\Omega}_j : \omega_j \sim \chi(0, \Omega_j)$.

$\boldsymbol{H}_j$ is the observation model which maps the true state space into the observing space and $\varepsilon_j$ is the observation noise which is assumed to be zero-mean Gaussian white noise with covariance $\boldsymbol{R}_j : \varepsilon_j \sim \chi(0, R_j)$.

Here is given a recursive process. We estimate the state at time $j$ through the state at time $j - 1$, then calculate the error correlation matrix $\boldsymbol{P}$ and Kalman gain $\boldsymbol{K}$, update the state variable, and then output $y$. The recursive algorithm of Kalman Filter is demonstrated as follows.

First, set both the $\boldsymbol{F}_1$ and $\boldsymbol{H}_1$ to be an identity matrix as the entrance of the recursive algorithm. Then, calculate each state $x$ at time $j$ predicted by state $x$ at time $j - 1$ defined as Equation (17):

$$
\overset{\wedge}{x}_{j|j-1} = \boldsymbol{F}_j x_{j-1|j-1}
\tag{17}
$$

Calculate the error covariance matrix that is between predicted and true values defined as Equation (18):

$$
\overset{\wedge}{\boldsymbol{P}}_{j|j-1} = \boldsymbol{F}_j \boldsymbol{P}_{j-1|j-1} \boldsymbol{F}_j^T + \boldsymbol{\Omega}_j
\tag{18}
$$

According to Equation (18), Kalman gain can be computed as Equation (19):

$$
\boldsymbol{K}_j = \overset{\wedge}{\boldsymbol{P}}_{j|j-1} \boldsymbol{H}_j^T \left( \boldsymbol{R}_j + \boldsymbol{H}_j \overset{\wedge}{\boldsymbol{P}}_{j|j-1} \boldsymbol{H}_j^T \right)^{-1}
\tag{19}
$$

Equations (17) and (19) are used to calculate the estimated value of $x$ at time $j$, defined as Equation (20):

$$
x_{j|j} = \overset{\wedge}{x}_{j|j-1} + \boldsymbol{K}_j \left( y_j - \boldsymbol{H}_j \overset{\wedge}{x}_{j|j-1} \right)
\tag{20}
$$

Then, compute the error covariance matrix that is between estimated and true values defined as Equation (21):

$$P_{j|j} = (I - K_j H_j) P_{j|j-1} (I - K_j H_j)^T + K_j R_j K_j^T \tag{21}$$

where $I$ is the identity matrix.

To obtain the optimal Kalman gain, the error covariance matrix further simplifies to Equation (22):

$$P_{j|j} = (I - K_j H_j) \hat{P}_{j|j-1} \tag{22}$$

Normally, Kalman Filter is more effective for suppressing Gaussian distributed noise. Otherwise, the robustness will be impacted. Therefore, we try to use the Huber-M estimation described in Section 2.2 to reconstitute the observables for improving the robustness.

The relationship between state truth value $x_j$ and its predicted value $\hat{x}_{j|j-1}$ at time $j$ is shown as Equation (23):

$$x_j = \hat{x}_{j|j-1} + \delta_j \tag{23}$$

where $\delta$ is the error of prediction. A linear regression model is constructed by combining Equation (16) as Equation (24):

$$\begin{bmatrix} y_j \\ \hat{x}_{j|j-1} \end{bmatrix} = \begin{bmatrix} H_j \\ I \end{bmatrix} x_j + \begin{bmatrix} \varepsilon_j \\ -\delta_j \end{bmatrix} \tag{24}$$

Then, we define Equation (25):

$$
\begin{aligned}
D_j &= \begin{bmatrix} R_j & 0 \\ 0 & P_{j|j-1} \end{bmatrix} \\
z_j &= D_j^{-1/2} \begin{bmatrix} y_j \\ \hat{x}_{j|j-1} \end{bmatrix} \\
M_j &= D_j^{-1/2} \begin{bmatrix} H_j \\ I \end{bmatrix} \\
\gamma_j &= D_j^{-1/2} \begin{bmatrix} \varepsilon_j \\ -\delta_j \end{bmatrix}
\end{aligned} \tag{25}
$$

Combined with Equation (2), we can define Equation (26):

$$z_j = M_j x_j + \gamma_j \tag{26}$$

By combining Equation (13), iteration $x_j$ is solved as described in Equation (27):

$$M_j^T W (z_j - M_j x_j) = 0 \tag{27}$$

where $M_j^T$ is the transpose of $M_j$, $W$ is the matrix of weight, and $z_j$ is the observation for true value $xj$.

The result of the $i + 1$ th iteration as Equation (28):

$$x_j^{i+1} = \left( M_j^T W M_j \right)^{-1} M_j^T W z_j \tag{28}$$

At the end of the iteration, the variance is obtained as Equation (29):

$$\hat{P}_{j|j} = \left( M_j^T W M_j \right)^{-1} \tag{29}$$

Introducing Equations (28) and (29) above results into the updating of the observations in Equations (20) and (22), and the Huber-M estimation Robust Kalman Filter algorithm is computed.

## 3. Results

### 3.1. Experiment Campaign

To validate the proposed models, we chose to use collected ground thrust data from Lamasquère area (Southwest of France, see Figure 2). The GNSS dataset was collected at Lamasquère, France (43°29′14.45″N, 1°13′44.11″) from 5th of February 2014 to 15th of March 2014. The details of the experiment campaign were thoroughly demonstrated in article [22], so in this study we only brief the key information. The site is a soya field equipped with a Leica GR25 multifrequency receiver and an AR10 antenna. At the same time, two Theta Probe ML3 soil moisture sensors are installed to collect ground truth measurements. The sensors measured soil moisture every 2 min at 2-cm and 5-cm depths with few meters away from the receiver. The antenna height is about 1.7 m. The soil type is silt clay, which consists of 18% sand, 41% clay and 41% silt. During dry conditions L-band waves do not penetrate more than a few centimeters (maximum observed 5.2 cm). The soil temperature varies between 5 °C and 16 °C. During the experiment campaign, the field had no vegetation cover and was considered as bare soil.

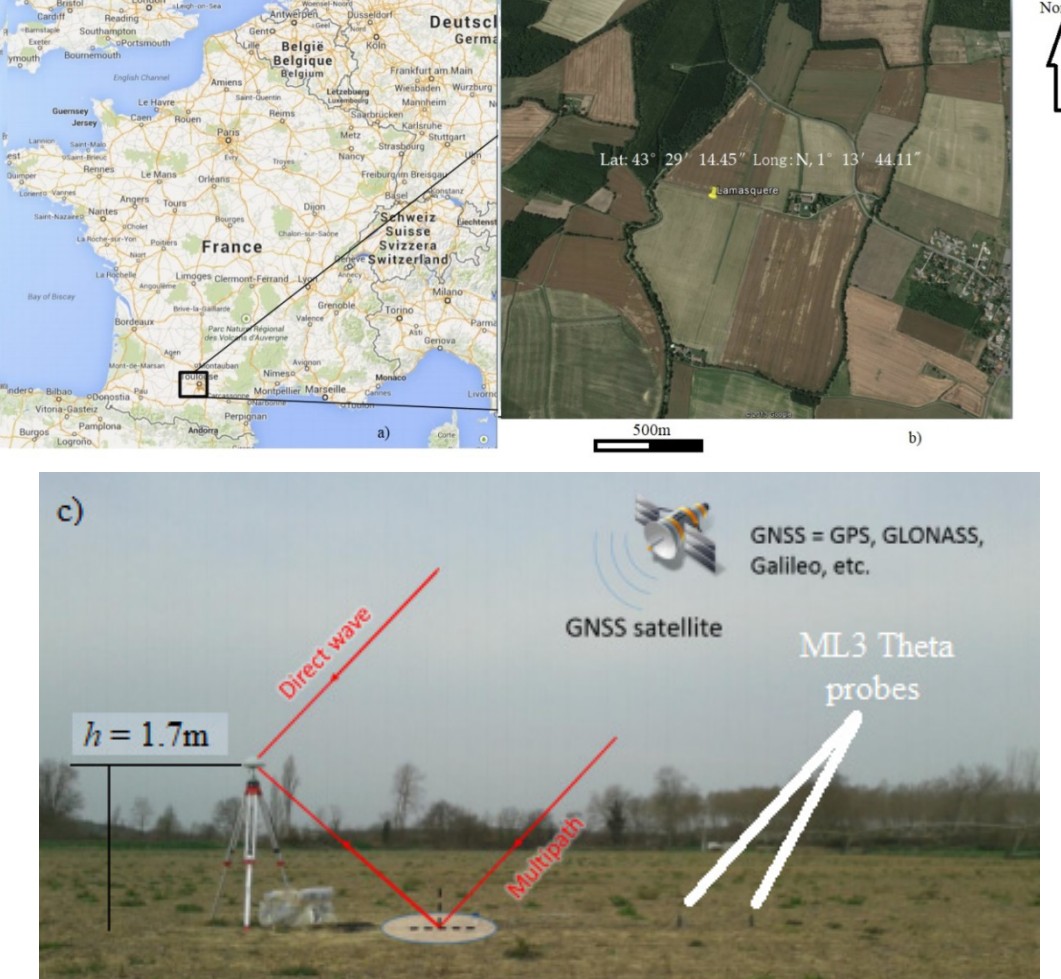

**Figure 2.** Lamasquère Experiment location: (**a**) map of France with a black box locating the region surrounding Lamasquère; (**b**) zoom on the Lamaquère plots where the measurements were taken (yellow dot); (**c**) zoom on the installed instrumentation with 2 ML3 Theta probes and Leica GR25 receiver and AR10 antenna.

In this study, the SNR data used for the validation came only from low-elevation satellites with angles ranging from 2° to 30° of elevation. First, the data was processed by the method mentioned in Section 2.1 to obtain the observables such as frequency, *h*, amplitude, and phase of SNR multipath oscillation. The second step was solving Equation (5) for the classical model and Equation (14) for the robust regression model. Then, the conventional regression model, robust regression model and the Robust Kalman Filter model were applied on the observables. For making a thorough comparison of those three models, including the conventional model, robust regression model and Robust Kalman Filter model with in situ measurement corresponding to the ground thrust data used for validation, we applied them using three scenarios:

(1) Single-band univariate regression, by using only one observable.

(2) Dual-band data fusion univariate regression.

(3) Dual-band data fusion multivariate regression

The data processing flow is demonstrated by Figure 3.

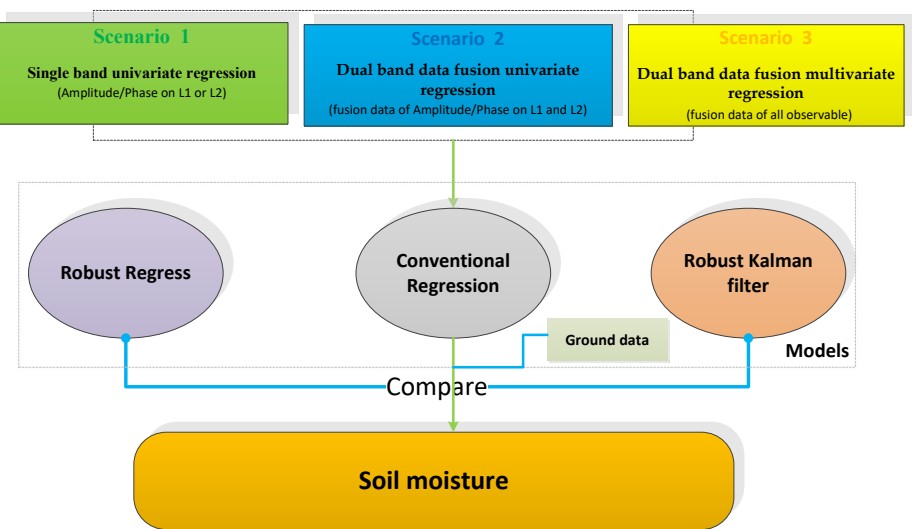

**Figure 3.** Data processing workflow: the in situ measurements of soil moisture are used as reference and are compared to the robust regression model and the Robust Kalman Filter model.

### 3.2. Comparison of Results of Single-Band Univariate Regression on GPS L1—Scenario One

In this section, we inverted the soil moisture by conventional linear regression model, robust regression model and robust Kalman model by using one of two observables, such as amplitude and phase. The comparisons of the results are shown as follows.

### 3.2.1. Using Amplitude Observables

First, we demonstrated the relationship between in situ soil moisture and amplitude observables by the making of PRN1 as an example in Figure 4.

We computed the correlation coefficients between the observables of amplitude and in situ soil moisture on L1 and L2 bands, as shown in Figure 4, where $R_{A1}$ is the correlation coefficients of amplitude on L1, and $R_{A2}$ is the correlation coefficients of amplitude on L2. It is clear to see that there is a linear correlation between amplitude and in situ soil moisture. For this satellite the correlation on the L2 band is better than L1. After training and predicting, we obtained the inversion results of the amplitude observable on GPS L2. We show the soil moistures retrievals in Figure 5 of our time series (47 days) for satellite PRN1 using the amplitude observable as input of the inverting model.

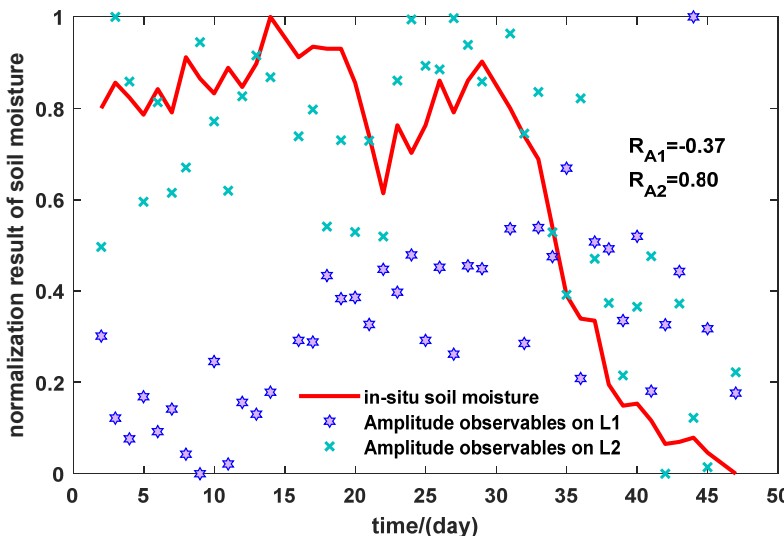

**Figure 4.** Normalization of in situ soil moisture and observables (amplitude of PRN1).

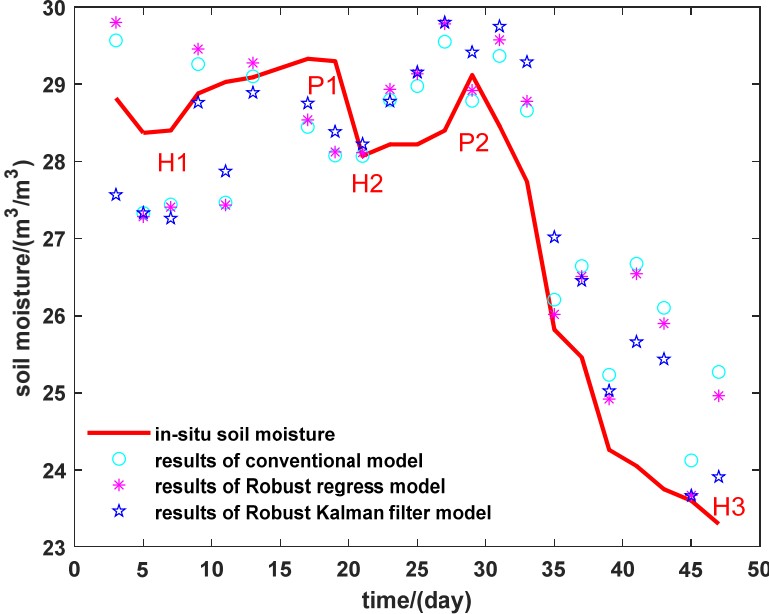

**Figure 5.** Estimation results of methods by using amplitude on L2 (PRN1), we highlighted for validation data the main moisture peaks (P1 day 17; P2 day 29) and holes (H1 day 6; H2 day 20; and H3 day 47). We can note that the difference between H1 and P1 is less than $1*10^{-2}$ $m^3/m^3$ and between H2 and P2 it is ~$1*10^{-2}$ $m^3/m^3$ which is low and will allow us to test the precision of the inversions and to show the capacity of this technique to differentiate between very weak soil moisture fluctuations.

We compared the results of the three models by using the amplitude observable with in situ soil moisture as our reference dataset. For the initialization step of all methods the first solution is coarse in comparison with the reference dataset. During days 5 until 20 one can see a gradual increase of the soil moisture for in situ data (from ~$28*10^{-2}$ $m^3/m^3$ to ~$29*10^{-2}$ $m^3/m^3$). As demonstrated in Figure 5, the results of the Robust Kalman Filter model show much better agreement to the increase of in situ soil moisture after day 10 than the other two models. The second increase (days 20 to 30) of the in situ soil moisture is clearly identified by the models (lower amplitudes, Figure 5). After day 35 the models differ more from real measurements. However, we can see clearly that the Robust Kalman Filter model improved the estimation results and is better than robust/classical regressions.

### 3.2.2. Using Phase Observables

Figure 6 demonstrates the relationship between observed phase and in situ soil moisture.

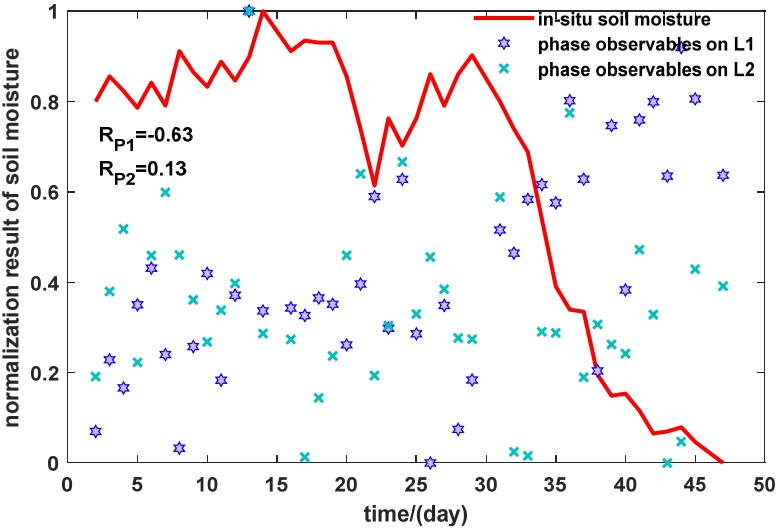

**Figure 6.** Normalization of in situ soil moisture and observables (phase on L1 for PRN1).

We computed the correlation coefficients between the observables of amplitude and in situ soil moisture on L1 and L2 bands, as Figure 6 shows, where $R_{P1}$ is the correlation coefficients of amplitude on L1, $R_{P2}$ is the correlation coefficients of amplitude on L2. According to Figure 6, when the in situ soil moisture increased, the observed phase on L1 and L2 showed a trend of decrease and vice versa, and both of them have a negative correlation. The inversion results of soil moisture using GPS L1 phase observable are shown in Figure 7.

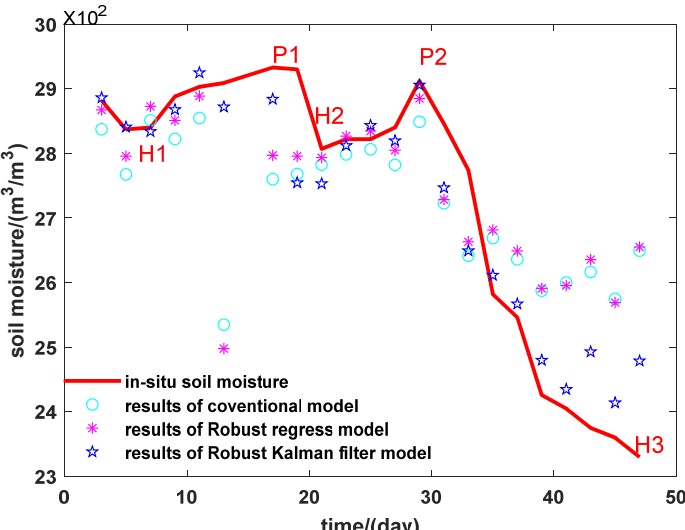

**Figure 7.** Estimation results of methods by using phase on L1 (PRN1), we highlighted for validation data the main moisture peaks (P1 day 17; P2 day 29) and holes (H1 day 6; H2 day 20; and H3 day 47). We can note that the difference between H1 and P1 is less than $1*10^{-2}$ m$^3$/m$^3$ and between H2 and P2 it is ~$1*10^{-2}$ m$^3$/m$^3$ which is low and will allow us to test the precision of the inversions and to show the capacity of this technique to differentiate between very weak soil moisture fluctuations.

We compared the soil moisture results of all the three models using phase with in situ soil moisture. The in situ time series changes were well reconstructed by the three models,

therefore the observed data in the first 10 days are well transcribed by the three models. We then have a sudden decrease until day 13 for the two linear regression models which show significant differences with in situ soil moisture. Until day 23 the Robust Kalman model continue to decrease which is more consistent with in situ data. For the three models, at day 30 they are all well reconstructed, especially the robust regression model, and the main decrease from day 30 to 45 is only visible with Robust Kalman model. The regression models are strongly divergent from the field data after day 35. The Robust Kalman model improves more significantly than the robust regression and classical ones.

We now compare phase results to those using amplitude as input. The results of the phase give better correlation and lower difference with respect to in situ soil moisture. As with the amplitude, the most robust model remains the one that uses the Kalman Filter, as it clearly follows the validation time series and presents small deviations from in situ time series even after day 37. The regression models, whether classic or robust, do not show this strong final decrease; we can see, however, that in using the phase the deviations from the validation data is smaller than for amplitude.

In general, in the single-band univariate scenario, compared with conventional method and the robust regression model, the Robust Kalman model can improve the precision of soil moisture inversion, and the Robust Kalman Filter model achieves much higher performance than regression models.

### 3.3. Comparison between Dual-Band Data Fusion Classical Regression Model, Robust Regression Model and Robust Kalman Model—Scenario Two

In scenario two, we invert the soil moisture using dual-band data fusion methods. For the four observables, amplitude and phase of both L1 and L2 bands, the comparisons of the results are shown as follows.

### 3.3.1. Using Amplitude

Figure 8 shows the results using the amplitude observable on GPS L1 and L2.

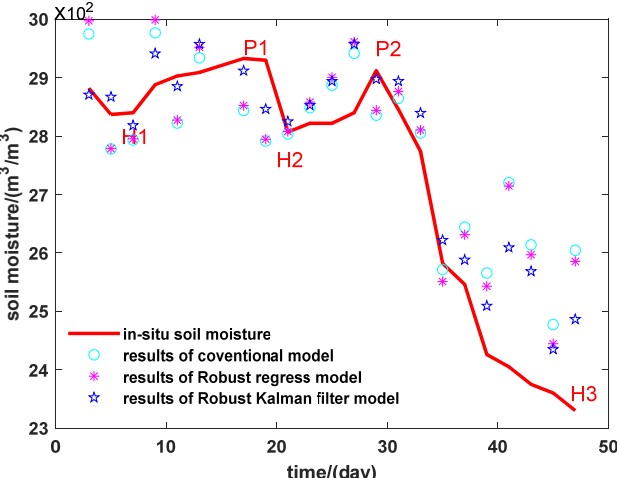

**Figure 8.** Estimation results of dual-band data fusion methods by using amplitude (PRN1), we highlighted for validation data the main moisture peaks (P1 day 17; P2 day 29) and holes (H1 day 6; H2 day 20; and H3 day 47). We can note that the difference between H1 and P1 is less than $1*10^{-2}$ m$^3$/m$^3$ and between H2 and P2 it is ~$1*10^{-2}$ m$^3$/m$^3$ which is low and will allow us to test the precision of the inversions and to show the capacity of this technique to differentiate between very weak soil moisture fluctuations.

We compared the soil moisture results of all the three models with dual-band using amplitude with in situ soil moisture. The in-situ time series changes are well reconstructed by the three models, therefore the observed data in the first 5 days are well transcribed by the Robust Kalman Filter models, while the other two regression models have a strange



difference. We then have a gradual increase from day 5 to day 20 with in situ soil moisture. The Robust Kalman Filter model has the lowest difference with in situ data but has an oscillation which is not in agreement with the decreasing trend of in situ soil moisture from day 40 to 47, while the other two models have more difference. The second increase (days 20 to 30) of the in situ soil moisture is clearly identified by all the models, and after day 35 the two models (robust regression and conventional regression) strongly differ from real measurements. However, we can clearly see that the Robust Kalman Filter model improved the estimation results better than the robust/classical regressions.

### 3.3.2. Using Phase

The inversion results of dual-band data fusion methods by using amplitude observables on GPS L1 and L2 are demonstrated in Figure 9.

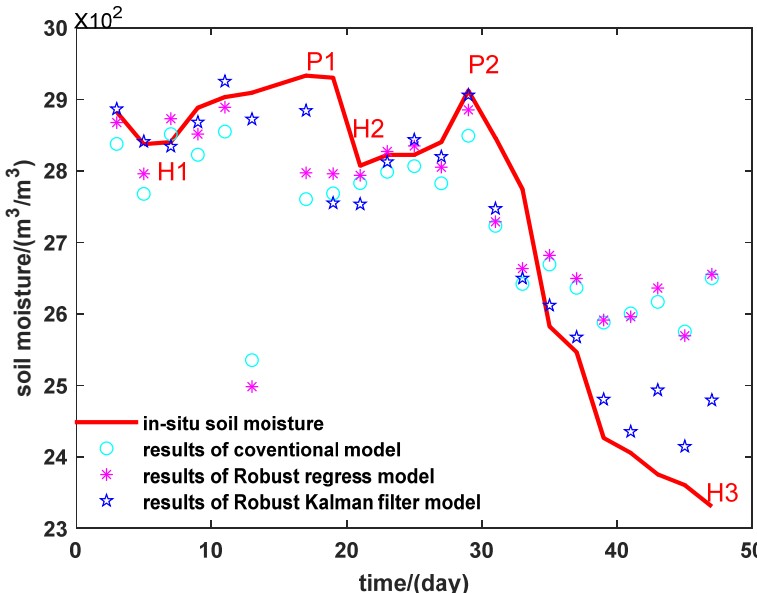

**Figure 9.** Estimation results of dual-band data fusion methods by using phase, we highlighted for validation data the main moisture peaks (P1 day 17; P2 day 29) and holes (H1 day 6; H2 day 20; and H3 day 47). We can note that the difference between H1 and P1 is less than $1*10^{-2}$ m$^3$/m$^3$ and between H2 and P2 it is ~$1*10^{-2}$ m$^3$/m$^3$ which is low and will allow us to test the precision of the inversions and to show the capacity of this technique to differentiate between very weak soil moisture fluctuations.

We compared the results of the three models by using the phase observable with in situ soil moisture as our reference dataset. The first 5 days for three models are more precise than the single-band scenario. During days 5 until 20 one can see a gradual increase of the soil moisture for in situ data (from ~$28*10^{-2}$ m$^3$/m$^3$ to ~$29*10^{-2}$ m$^3$/m$^3$). For the two models (robust regression and conventional regression) in general this growth has a dramatic reduction on day 13. The second increase (days 20 to 30) as compared to scenario one of the in situ soil moisture is clearly identified by the models. The robust regression model has a lower difference in general regarding this growth, and after day 30 the models strongly decrease. However, we can see clearly that the Robust Kalman Filter model improved the estimation results better than the robust/classical regressions. The two linear regression models differ with in situ soil moisture from day 35, but the Robust Kalman Filter model improved the estimation results and agreed with the decreasing trend.

In general, for the dual-band data fusion scenario, compared with the conventional method and robust regression method, the Robust Kalman model can improve the precision of soil moisture inversion when compared with the data fusion conventional method and robust regression method. Meanwhile, the data fusion Robust Kalman improves more

obviously with all the observables than the robust regression method if comparing them in a single-band scenario.

### 3.4. Multivariate Variable Dual-Band Data Fusion—Scenario Three

In this section we invert the soil moisture by conventional linear regression model, robust regression model and Robust Kalman model by using all of the observables on both GPS L1 and L2 bands simultaneously. The comparison results are shown in Figure 10.

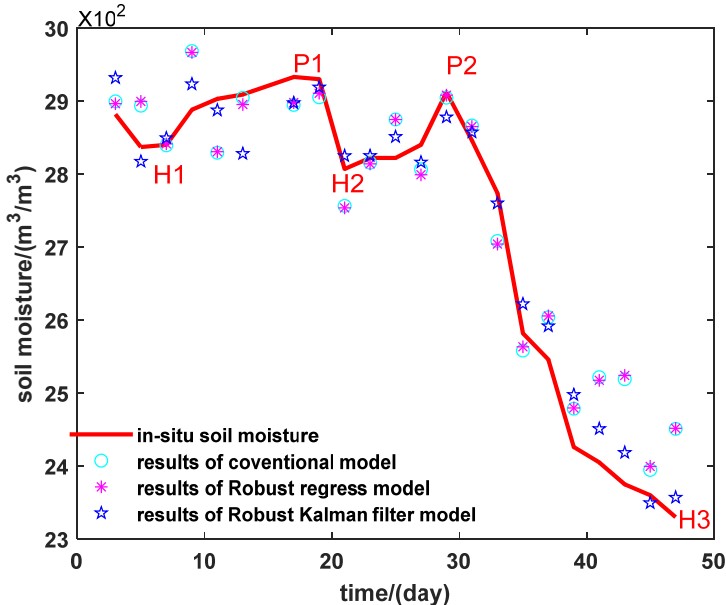

**Figure 10.** Estimation results of multivariate variable dual-band data fusion methods (PRN1), we highlighted for validation data the main moisture peaks (P1 day 17; P2 day 29) and holes (H1 day 6; H2 day 20; and H3 day 47). We can note that the difference between H1 and P1 is less than $1*10^{-2}$ m$^3$/m$^3$ and between H2 and P2 it is ~$1*10^{-2}$ m$^3$/m$^3$ which is low and will allow us to test the precision of the inversions and to show the capacity of this technique to differentiate between very weak soil moisture fluctuations.

We compared the results of the three models by using multivariate variable dual-band with in situ soil moisture as our reference dataset. In the first 5 days, the Robust Kalman Filter model shows minimal differences with in situ data, and it is better than scenario one and two. The other two models have much more differences with in situ data. During days 5 until 20 one can see a gradual increase of the soil moisture for in situ data (from ~$28*10^{-2}$ m$^3$/m$^3$ to ~$29*10^{-2}$ m$^3$/m$^3$). For the Robust Kalman Filter model in general this growth is much faster and reaches a maximum intensity after 10 days. The second increase (days 20 to 30) of the in situ soil moisture is clearly identified by all the three models, and after day 35 models of conventional method and robust regression method differ from real measurements, but they are better than the other two scenarios. However, we can see clearly that the Robust Kalman Filter model improved the estimation results better than the robust/classical regressions, and it is also better than the previous scenarios.

### 4. Analysis and Discussion

It has been demonstrated that GNSS-IR has the ability to retrieve land surface parameters, and especially the observable of soil moisture. After K. Larson first retrieved soil moisture from GPS SNR data [14], massive studies have developed models for GNSS-IR soil moisture inversion. Although most studies focus on low elevation angles, in article [37] which was mentioned above, the authors take the pseudo-dynamic of the surface into account and obtain a significantly improved and also utilized signal with high elevation

angles. How to apply the Robust Kalman Filter model with signal of high elevation angles is a main interest in future work.

In the experimental campaign of Lamasquère, the land is bare soil, but how to eliminate the impact of vegetation is also important. In article [38], the author proposed a multivariate adaptive regression spline method, considering the impact of the vegetation moisture content, with a correlation coefficient of 0.916 and a root-mean-squared error of 0.021 $m^3/m^3$. In article [39], the authors performed a 15-month observation which covered an entire growing cycle by two antennas and developed an inversion model on GPS L2C and L5 SNR, achieving a precision of 0.035 $m^3/m^3$ for the whole meadow growing cycle, and of 0.018 $m^3/m^3$ after grass cutting.

For every model, the correlation coefficients are calculated for each satellite on each band or dual-band combination as Figure 11. Satellites in which correlation coefficients are over 0.5 are considered as effective cases.

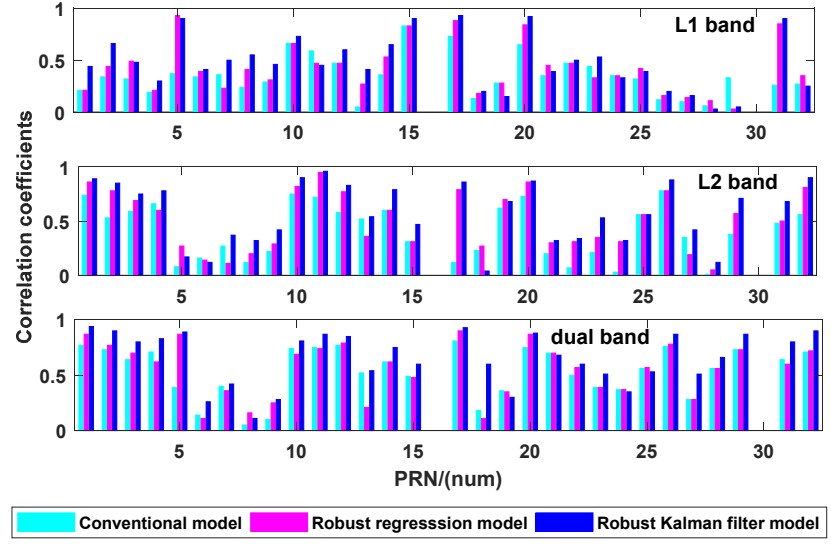

**Figure 11.** Correlation coefficients of three methods on each frequency with amplitude.

Compared with the conventional model, the robust regression model increases on average 28.77% for L1, 18.33% for L2 and 5% for dual-band. The correlation coefficient of the Robust Kalman Filter model increases on average 32.33% for L1, 28.14% for L2 and 19.10% for dual-band. As the statistical data shows, the Robust Kalman Filter model achieves the highest precision.

The correlation coefficients of the three models show that the Robust Kalman Filter model gives a better correlation than regression models, except for satellites PRN 6, 7, 8, 9, 19 and 24, for which the correlation coefficient is very weak, reflecting a non-correlation (R < 0.5). For demonstration, we counted the number of effective satellites for all the methods, as shown in Table 1.

**Table 1.** Number of effective satellites used amplitude on each band.

| Method | L1 | L2 | Dual-Band |
| --- | --- | --- | --- |
| Conventional model | 4 | 14 | 19 |
| Robust regression model | 7 | 16 | 19 |
| Robust Kalman model | 13 | 18 | 24 |

There are 7 satellites for the robust regression model and 4 for conventional regression model on the L1 band, but the Robust Kalman Filter model has 13 satellites for soil moisture inversion. On the other hand, data fusion has also had a positive impact on the increase in the number of effective satellites. If we compare the results of the conventional method

for dual-band and single-band, the correlation coefficient of the L1 band with an average increment is 48.66% and a maximum grow of 90.38%. Meanwhile, the correlation coefficient of L2 with an average increment is 26.93% and a maximum grow of 98.21%. For the robust regression model, the correlation coefficient of the L1 band with an average increment is 34.45% and a maximum grow of 80.36%. Meanwhile the correlation coefficient of L2 with an average increment is 14.06% and a maximum grow of 91.07%. The correlation coefficient of the Robust Kalman Filter model on the L1 band with an average increment is 34.80% and a maximum grow of 95.45%. Meanwhile, the correlation coefficient of L2 with an average increment is 18.50% and a maximum grow of 93.33%.

Then, we analyzed the correlation between the phase observable and in situ soil moisture as shown in Figure 12.

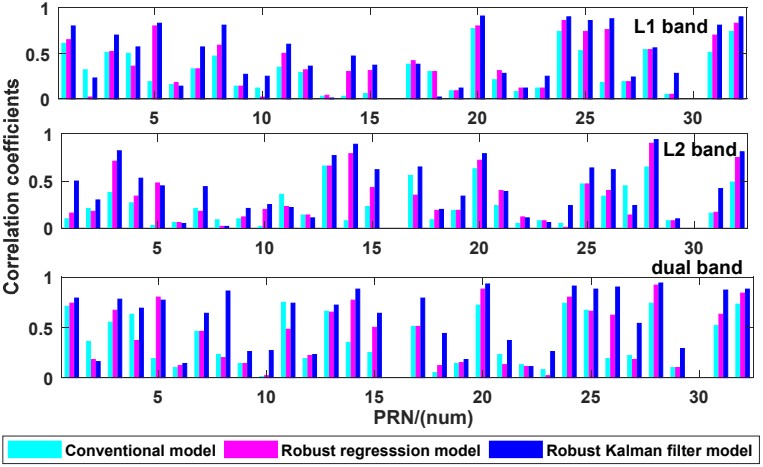

**Figure 12.** Correlation coefficients of three methods on each frequency with phase.

The correlation of phase is not better than amplitude and there are few satellites effective for soil moisture estimation. However, we can see that there are some satellites which are not effective for conventional methods but effective for robust regression and Robust Kalman Filter method. It shows both the robust regression and Robust Kalman Filter models are capable of increasing the number of effective satellites, such as PRN5 and PRN26 for the L1 band, and PRN3 and PRN14 for the L2 band. The figure also shows that the dual-band data fusion method makes improvement of correlation between the phase observable and in situ soil moisture.

Compared with the conventional model, the robust regression model increased by an average of 24.59% for L1, 35.22% for L2 and 24.63% for dual-band. The correlation coefficient of the Robust Kalman Filter model increased by an average of 33.96% on L1, 43.92% on L2 and 35.29% for dual-band. The Robust Kalman Filter can greatly improve the inversion accuracy of the model regardless of single frequency or dual-band fusion.

If we look the precision of the three models, i.e., correlation coefficients, one can see that the Robust Kalman Filter model gives a better correlation than regression models, except for satellites PRN 6, 9, 10, 12, 18, 19, 21, 22, 23 and 29, where the correlation coefficient is very weak and reflects a non-correlation (R < 0.5). For demonstration, we counted the number of effective satellites for all the methods, as shown Table 2.

**Table 2.** Number of effective satellites used phase on each band.

| Method | L1 | L2 | Dual-Band |
|---|---|---|---|
| Conventional model | 9 | 4 | 12 |
| Robust regression model | 12 | 6 | 14 |
| Robust Kalman model | 14 | 12 | 18 |

The table shows that there are more effective satellites on the L1 band than L2 band and the dual-band data fusion also has a positive impact for increasing the number of effective satellites.

If we compare the results of the conventional method for dual-band and single-band, the correlation coefficient of the L1 band with an average increment is 25.68% and a maximum grow of 95.45%. Meanwhile, the correlation coefficient of L2 with an average increment is 36.46% for L2 and a maximum grow of 93.24%. For the robust regression model, the correlation coefficient of the L1 band with an average increment is 17.38% and a maximum grow of 93.84%. Meanwhile, the correlation coefficient of L2 with an average increment is 28.60% and a maximum grow of 98.75%. The correlation coefficient of the Robust Kalman Filter model on the L1 band with an average increment is 19.74% and a maximum grow of 98.61%. Meanwhile, the correlation coefficient of L2 with an average increment is 31.76% and a maximum grow of 97.67%.

Compared with the correlation coefficient of the amplitude observable, dual-frequency fusion has a better effect on the L2 band and a greater improvement. However, the amplitude is reversed.

For every model, the root-mean-squared error (RMSE) is also calculated for each satellite on each band or dual band combination. The results are shown in Figures 13 and 14. All the discussion for RMSE is based on valid satellites defined in line 498~499.

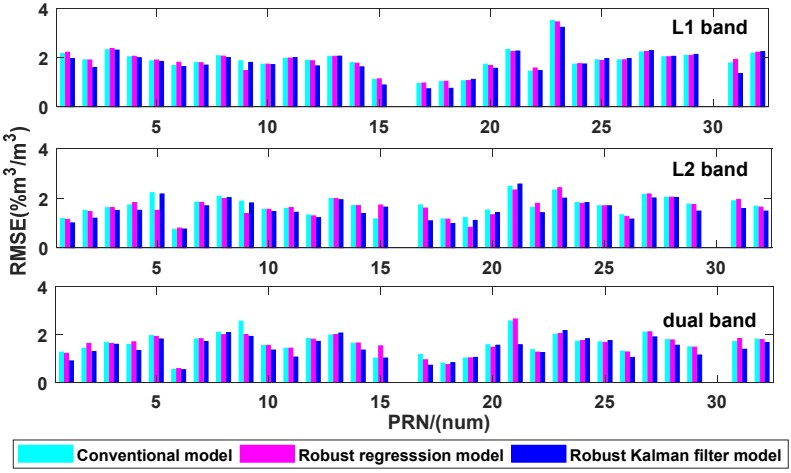

**Figure 13.** RMSE of three methods on each frequency with amplitude.

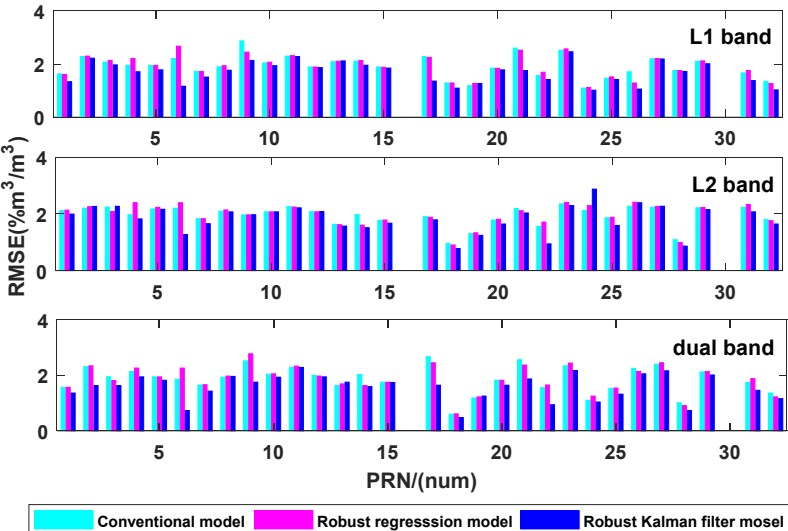

**Figure 14.** RMSE of three methods on each frequency with phase.

First, we discuss the results with amplitude. Compared to the conventional method, the RMSE of the robust regression model decreases by an average of 2.54% on L1 and 2.81% on L2, meanwhile the RMSE of the Robust Kalman Filter model decreases by an average of 10.19% on L1 and 12.39% on L2. Over half of the satellites' RMSE are between 1% m$^3$/m$^3$ and 2% m$^3$/m$^3$ for the Robust Kalman Filter model.

After dual-band data fusion, the RMSE of the conventional model decreases 12.10% more than L1 and 3.46% more than L2. The RMSE of robust regression model decreases 14.30% more than L1 and 0.72% more than L2. The RMSE of the Robust Kalman Filter model decreases 18.23% more than L1 and 7.28% more than L2. As the statistical data show, the Robust Kalman Filter model demonstrates the highest precision.

Moreover, we discuss the results with phase. Compared to the conventional method, the RMSE of the robust regression model decreases by an average of 1.32% on L1 and 6.11% on L2, meanwhile the RMSE of the Robust Kalman Filter model decreases by an average of 11.49% on L1 and 8.87% on L2. Over half of satellites' RMSE are between 1% m$^3$/m$^3$ and 2% m$^3$/m$^3$ for the Robust Kalman Filter model.

After dual-band data fusion, the RMSE of conventional model decreases 5.45% more than L1 and 46.5% more than L2. The RMSE of the robust regression model decreases 2.08% more than L1 and 45.01% more than L2. The RMSE of the Robust Kalman Filter model decreases 1.24% more than L1 and 47.18% more than L2. As the statistical data show, the Robust Kalman Filter model demonstrates the highest precision.

In the end, we discuss dual-band multivariate data fusion models. The correlation coefficients and RMSE are shown in Figure 15.

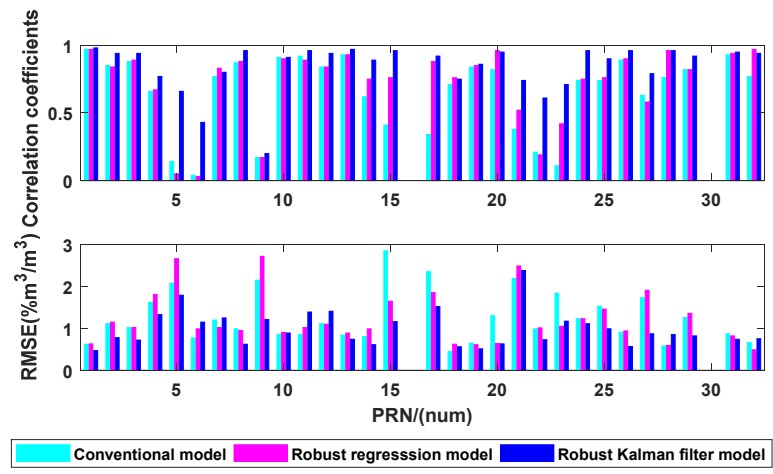

**Figure 15.** Correlation coefficients and RMSE of dual-band with multivariate.

As demonstrated in Figure 15, the correlation between the inverted and the in situ soil moisture has been significantly enhanced for all the models, regardless of single-band or dual-band data fusion approach. The conventional model has 22 effective satellites un-der the multivariate fusion scenario, while the robust regression and Robust Kalman Filter methods have 25 and 28 satellites, respectively, by which means we have more effective satellites to estimate soil moisture than univariate models.

Compared with the results of the univariate models, the correlation coefficient of dual-band data fusion multivariate conventional method self-improves for all the GPS satellites with an average increase of 44.97% for phase and an average increase of 29.43% for amplitude; meanwhile, the RMSE decreases by an average of 37.00% for amplitude. The correlation coefficient of dual-band data fusion multivariate robust regression model self-improves with an average increase of 29.43%, meanwhile the RMSE decreases for all satellites with an average reduction of 29.42% for amplitude. The correlation coefficient of dual-band data fusion multivariate Robust Kalman Filter self-improves with an average increase of 20.7%, meanwhile the RMSE decreases for all satellites with an average reduction of 31.42%.

## 5. Conclusions

The inversion models of GNSS-IR soil moisture remote sensing have been investigated in this paper. First, we introduced the conventional regression model and the robust regression model and proposed a Robust Kalman Filter model. Then, the validation of the models was carried out in three different scenarios, including the "single-band, single-observable" scenario, the "dual-band, single-observable" scenario and the "dual-band, dual-observables" scenario. In the single-observable scenarios, the results retrieved using the phase observable had better performance than those retrieved using the amplitude observable, and dual-band data fusion performed better than single-band retrieval. For the dual-band scenarios, the results obtained using dual observables achieved better performance than those retrieved using a single observable. Furthermore, the results provided by the proposed model achieved superior performance to those obtained by the other two models in each scenario. This indicated that the proposed model was available for the data fusion of both dual-band and dual-observable cases, which was significant to take full use of the multi observables provided by multi-band GNSS signals to improve the retrieval performance in GNSS-IR soil moisture remote sensing.

**Author Contributions:** Conceptualization, L.J., F.G. and L.Y.; methodology, L.Y. and D.Y.; software, L.J., W.Y., H.R. and X.H.; data collection, J.D.; validation, L.J., B.S., W.Y. and Y.L.; writing—original draft preparation, L.Y. and L.J.; writing—review and editing, T.X., X.H., J.D. and N.W.; project administration, L.Y.; funding acquisition, L.Y. and H.R. All authors have read and agreed to the published version of the manuscript.

**Funding:** This research was funded by National Natural Science Foundation of China, grant number 31971781, and Zhejiang Provincial Basic Public Welfare Research Project Foundation of China, grant number LGN19D040001.

**Acknowledgments:** The authors would like to thank Nicolas Roussel for data collection and Mutian Han for the help of data processing.

**Conflicts of Interest:** The authors declare no conflict of interest.

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
