# Peer review of "Robust Kalman Filter Soil Moisture Inversion Model Using GPS SNR Data—A Dual-Band Data Fusion Approach"

_remotesensing, doi:10.3390/rs13194013_

Round 1
Reviewer 1 Report
The paper deals with the determination of the correlation coefficient between derived SNR observations from GNSS and soil moisture in-situ observations.
The paper is interesting and works on a relatively novel area, however some modifications should be made before publication:
- I think the principal novelty of the paper is missing: the main problem of the “classical method”, that is, the use of a value of 65.1°/(m3/m3) for this linear relationship in case of use phase determinations, or a better slope computed by adjusted the satellite tracks with in situ soil moisture variations, is the reduction of the number of valid satellites that can be used in the final soil moisture monitoring because of the low correlation some satellites present with respect to the reference values. I think that the principal novelty of the paper should be the attempt to increase this number of satellites that are finally used to monitoring soil moisture by GNSS-IR. In order to achieve this objective data fusion and the use of robust regression model and robust Kalman filter model are proposed.
So, I think authors, maybe, should rethink the real purpose of their manuscript contribution.
- In general, the paper is well writing, but some modifications are necessary to improve the reading: The discussion section is not a real “discussion section” where the results are discussed. My recommendation is to rewrite a new discussion section where all the text from sections 3.2, 3.3 and 3.4 just after the figures of satellites correlation and RMSE are included. That will end with a results section containing only results and discussion section containing the discussion on the results. However, this new discussion section should include tables where all numerical values for all methods are included to improve the understanding of the results. It is difficult to follow the thinking of the authors about the results as the manuscript is written, and I think this proposal with new tables will help readers.
- Figure 1 looks blurry, at least in the pdf file I am working with. There are two blue lines so I recommend changing the color of the reflected signal.
- Line 112, what do you mean by AKA?, could you please include this angle in figure 1?
- Line 122, please write the complete words “with refers to” instead the abbreviation w.r.t.
- Please be more precise with the notation of the formulas, for example in equation 2, X is the sine of the elevation, so X cannot be used for other purpose, for example in equation 3 or equation 5. In line 157, X is the dependent variable matrix, but in line 161 are observables…this happens throughout the entire text.
- In equation 10 all conditions are the same.
- Where the 0.6745 value of equation 11 comes from?
- What is f(uj) from equation 13?
- Is equation 14 written correctly?
- Lines 215-216. The end condition of the algorithm is not correct. The use of robust estimation can lead some weights to get very close to zero (completely eliminating some observables) and others to tend to infinity (oversizing their importance within the system of equations), to avoid this, it is normal to use a limited number of iterations, for example 10, where in the first three the robust estimation is not used so as to obtain results that are not biased by the relative importance of the observables.
- In the definition of the Kalman filter, what is the transition matrix Fj used (maybe the identity matrix), what are the values to include in the covariance matrix of the process noise (maybe zero)?
- Equations 30, 31, 32 and 33 are referenced but not included in the text (lines 280 and 281).
- Could you please include the amplitude observations in figures 4 and 10, and phase observations in figures 7 and 13?
- I assume that a different correlation coefficient is being calculated for phases, amplitudes, or a combination of both, but, is a different correlation coefficient being calculated for each satellite and for each frequency or is it the same for all satellites and for both frequencies? This is not clear from the text. In any case, a table should be included with all the correlation coefficients calculated and their variations explained appropriately.
- Line 457: four observables, not three.
-In the caption of Figure 6 (and the other similar) is the percentage symbol correct? I mean, I think this symbol has been included by mistake.
Reviewer 2 Report
- The authors are encouraged to add relevant citations after "... and atmosphere." (line 38).
- There are numerous, important and more recent GNSS-R soil moisture retrieval papers that ought to be cited in addition to [3] in line 48. Those include more recent works (on CYGNSS) by Dr. C. Chew as well as Al-Khaldi et al., Lakshmi et al. and Yueh et al.
- It is important to recognize and make clear within the manuscript that the GPS simulator developed by Chow may not be able to completely capture realistic GNSS-R behavior in response to varied levels of soil moisture and vegetation particularly for spaceborne GNSS-R observations. In fact, leading GNSS-R groups are only just starting to grapple with the various complexities of land GNSS-R reflections and developing complementary models.
- In the discussion starting with line 79, the authors seem to suggest that there had been a deficiency on other authors' side due to the fact that existing literature does not cover, in any extensive detail, dual band GNSS-R remote sensing (or at least multi-frequency GNSS-R remote sensing) and/or subsequently data fusion. It is important that the authors recognize and make clear that this is primarily due to the fact that flag ship GNSS-R platforms, TDS-1 and CYGNSS, do not have multi-frequency capability as part of their standard operations and are only capable of making standard measurements within at L1 (1.575 GHz).
- Refer to line 97: The Doppler shift can be ignored because the authors' appear to be using a ground based step up. The Doppler shift cannot and is not typically ignored for say, spaceborne GNSS-R receivers. It is important that these nuances are made clear within the body of the manuscript.
- Refer to line 203: When you say "s_j for each satellite is introduced" are you referring to the GNSS satellite(s)? I would assume so since the manuscript appears to be concerned with a ground based set up. If so, I suggest citing works by T. Wang et al. on GPS EIRP variability here which could provide an impetus for why satellite specific s_j estimation is necessary without the authors having to delve into too much detail.
- Do the authors have some sense of the reasons underlying the discrepancy at H3 in Fig. 4?
Round 2
Reviewer 1 Report
-Line 45. What is AKA in the text?
-Line 286. Equation (28) instead of equation (30).
-Figures 4 and 6. I do not see clearly the correlation, could you please include the correlation coefficient in figure captions?
- I think I was wrong in comment 14 made in the previous review. I was not referring to the correlation between in-situ soil moisture and predictions for every satellite (I understand that there is a different correlation for each satellite), but to the coefficients in the β matrix of coefficients (for example in equation 5): are different coefficients being calculated for each satellite and for each frequency or are they the same for all satellites and for both frequencies? This is not clear from the text. In any case, a table should be included with all the computed coefficients (especially the slope of the lineal regression) and their variations explained appropriately.
Reviewer 2 Report
The responses to this reviewer's initial concerns are satisfactory. I recommend accepting the manuscript.
Author Response
Dear reviewer,
Thank you for the advices on our manuscript.